# Transfer Learning for Alzheimer’s Disease through Neuroimaging Biomarkers: A Systematic Review

**DOI:** 10.3390/s21217259

**Published:** 2021-10-31

**Authors:** Deevyankar Agarwal, Gonçalo Marques, Isabel de la Torre-Díez, Manuel A. Franco Martin, Begoña García Zapiraín, Francisco Martín Rodríguez

**Affiliations:** 1Department of Signal Theory and Communications and Telematics Engineering, University of Valladolid, Paseo de Belén 15, 47011 Valladolid, Spain; goncalosantosmarques@gmail.com (G.M.); isator@tel.uva.es (I.d.l.T.-D.); 2Polytechnic of Coimbra, ESTGOH, Rua General Santos Costa, 3400-124 Oliveira do Hospital, Portugal; 3Psychiatric Department, University Rio Hortega Hospital–Valladolid, 47011 Valladolid, Spain; mfrancom@saludcastillayleon.es; 4eVIDA Laboratory, University of Deusto, Avenida de las Universidades 24, 48007 Bilbao, Spain; mbgarciazapi@deusto.es; 5Advanced Clinical Simulation Center, School of Medicine, University of Valladolid, 47011 Valladolid, Spain; fmartin@saludcastillayleon.es

**Keywords:** Alzheimer’s disease, neuroimaging biomarkers, magnetic resonance imaging, positron emission tomography, transfer learning

## Abstract

Alzheimer’s disease (AD) is a remarkable challenge for healthcare in the 21st century. Since 2017, deep learning models with transfer learning approaches have been gaining recognition in AD detection, and progression prediction by using neuroimaging biomarkers. This paper presents a systematic review of the current state of early AD detection by using deep learning models with transfer learning and neuroimaging biomarkers. Five databases were used and the results before screening report 215 studies published between 2010 and 2020. After screening, 13 studies met the inclusion criteria. We noted that the maximum accuracy achieved to date for AD classification is 98.20% by using the combination of 3D convolutional networks and local transfer learning, and that for the prognostic prediction of AD is 87.78% by using pre-trained 3D convolutional network-based architectures. The results show that transfer learning helps researchers in developing a more accurate system for the early diagnosis of AD. However, there is a need to consider some points in future research, such as improving the accuracy of the prognostic prediction of AD, exploring additional biomarkers such as tau-PET and amyloid-PET to understand highly discriminative feature representation to separate similar brain patterns, managing the size of the datasets due to the limited availability.

## 1. Introduction

The commonest form of dementia is Alzheimer’s disease (AD), with no authenticated disease-modifying treatment [1]. AD progresses gradually for several years before clinical symptoms [1]. It is estimated that approximately 5.8 million individuals in the USA were living with AD in 2020. This number is anticipated to be 14 million by 2050 [2]. It is critical to identify individuals at an initial stage in AD, even before meeting the clinical criteria. Mild cognitive impairment (MCI) characterizes an effort to distinguish patients at an early clinical phase, and has been considered as a target for clinical trials [3]. MCI is a stage between healthy and AD. A subject with MCI will have changes in their cognitive ability, but still will be able to perform their daily activities. Nearly 20% of individuals aged 65 or above have MCI. In total, 35% of them develop AD in 3 to 5 years [2]. MCI subjects will either be converted to AD or will remain stable. A confirmed diagnosis of Alzheimer’s disease (AD) can only be identified through autopsy [4].

Clinical experts are using magnetic resonance imaging (MRI) and positron emission tomography (PET) neuroimaging, neuronal injury, amyloid and tau biomarkers, cerebrospinal fluid, and degeneration [5,6,7] for AD diagnosis. Mini-Mental State Examination [8], and Clinical Dementia Rating [9] are the two recommended tests for AD diagnosis. The estimated cost of managing AD in the USA, comprising the communal, medical, and income loss to the patient family members in 2020, was approximately USD 305 billion. This cost is expected to rise as high as USD 1.1 trillion by 2050 [2]. The early detection of AD is critical before clinical manifestation for timely treatment. Therefore, a multi-class decision system to identify AD and its different stages from normal controls are required. Categorizing AD from MCI or normal patients is imperative, as AD is detectable without the aid of any experts or skills when it is too late for treatment. The main challenge is the progression prediction of MCI to AD and the distinction of MCI from normal controls in order to give timely treatment to patients. The progress made in neuroimaging techniques allowed physicians the opportunity to attempt to integrate highly dimensional multimodal neuroimaging data by using existing systems. Therefore, researchers recommended the use of computer-aided machine learning (ML) algorithms for AD diagnosis. ML pattern analysis methods that have performed well for AD detection include support vector machines, logistic regression, and support vector machine-recursive feature elimination [10]. Competitions such as Alzheimer’s disease Big Data Challenge [11] and the challenge of predicting MCI from MRI data [12] have revealed usefulness in the AD detection process by providing a common platform for researchers from all over the world. These competitions have produced several ML algorithms that have been implemented and assessed. To use these ML methods for classifications, the architectural design must be predefined. In general, four steps are required, namely the extraction of the features, the selection of the features, the reduction in the dimensions, and the implementation of a classification algorithm. Moreover, this process requires optimization and the involvement of experts in each stage [13]. The reproducibility of these implementations has been an issue [14].

The increasing capacity of GPUs enabled the spread of DL algorithms for image classification tasks. DL is the branch of ML that emulates the working of the human brain for identifying complex patterns. It uses raw neuroimaging data to learn features, hidden representation, and disease-related patterns, and explore associations in different parts of images through impulsive learning, which attracts researchers from the fields of highly dimensional medical image analysis, image segmentation, objects recognition, and disease detection [15]. On the one hand, DL models have been efficaciously applied to neuroimaging images such as MRI, functional MRI, PET, and diffusion-tensor imaging. On the other hand, DL methods, specifically convolutional neural networks(CNNs) have been shown to have better performance in comparison to the existing ML methods [13]. Recent studies suggest that transfer learning (TL) has turned out to be very popular in the domain of DL, as it enables DL training to be effective in the case of insufficient data [16,17]. TL is recommended by comparison with human behavior, since we can use knowledge learned in the previous tasks for solving new complex problems. Classifying AD and normal controls is simple when compared to classifying progressive MCI and stable MCI. Additionally, the amount of neuroimaging data for the classification of progressive MCI vs. stable MCI (pMCI vs. sMCI) is comparatively less as compared to the first task. However, both types of tasks shared common biomarkers [18]. Based on this perception, local TL has been implemented in many studies. The concept of local TL is to use the finalized weights of the AD vs. normal control classifier as the initial weights for the classification of sMCIvs. pMCI. Moreover, some included studies also used pre-trained 3D CNN-based architectures such as VGG16, ResNet for initializing the weights for the classification of AD vs. normal control, and local TL for initializing the weights for the classification of stable MCI vs. progressive MCI, and obtained better results. TL helps to speed up the training procedure and improve the performance of DL architectures [19].

The main objective of this study is to explore the neuroimaging biomarkers and their features, available databases, and pre-processing and input management techniques, the DL and TL methods that are adept for apprehending the disease-related patterns and perform efficiently even in the case of insufficient neuroimaging data, the accuracy achieved to date, the limitations, and gaps. Consequently, we systematically reviewed the papers of the last 10 years. These papers were reviewed, evaluated, and analyzed based on the DL and TL algorithms and neuroimaging biomarkers. Furthermore, we discussed the outcomes and gaps for DL applications in the AD research domain. This study aims to support future researchers that want to work in this field.

The rest of this article is structured as follows: Section 2 provides the research methodology, including search strategies, research questions, and inclusion and exclusion criteria. Section 3 provides the answers to research questions, along with results and a discussion, followed by the conclusion section.

## 2. Materials and Methods

A systematic literature review was carried out by the authors, considering studies published over the last ten years. The included studies focus on DL architectures, TL, neuroimaging data to classify AD, and predicting the transformation from MCI to AD. This study was conducted in different steps according to PRISMA methodology [20]. The main objective was to identify the gaps in the state of the art. Firstly, the research questions (RQs) were identified. Secondly, the search strategy was implemented, and the inclusion and exclusion criteria were identified for selecting the relevant articles. Finally, data extraction was carried out to answer the RQs. Furthermore, answers to these RQs were given while highlighting the challenges, limitations, and future scope in the field. These steps were explained in the following sub-sections, respectively.

### 2.1. Research Questions

To support future research activities, it is mandatory to examine the critical aspects of the current literature. The objective of this is to help develop an improved decision support system for the early diagnosis of AD. The following RQs were identified to represent critical aspects needed.

RQ 1: Which DL models and TL approaches have been used for capturing disease-related patterns?RQ 2: Which are the neuroimaging biomarkers and the parameters used?RQ 3: Which are the pre-processing techniques used to handle neuroimaging biomarkers?RQ 4: What is the current state of accuracy and other performance measures for the diagnostic classification and prognostic prediction of AD?RQ 5: How many and what structures use the opensource, available datasets?RQ 6: Which software platforms are needed for pre-processing neuroimaging data and implementing DL algorithms through a transfer learning approach?RQ 7: What are the techniques that have been used to reduce overfitting?RQ 8: What are the current research gaps, challenges, and opportunities?

### 2.2. Search Strategy

Science Direct, IEEE Xplore, Web of Science, ACM Digital Library, and PubMed were chosen to identify contributions in DL applications for the early diagnosis of AD by using TL in neuroimaging data. These databases were selected since they include the most relevant peer-reviewed studies on DL and are the more reputable sources. The search process was carried out by using the following query:

“Deep Learning” AND “Alzheimer” AND “neuroimaging” AND “Transfer Learning”

The first author performed the search by using Sysrev [21]. This is a platform for the collaborative extraction of data from documents available online that provides support by building article filters through ML. In Figure 1, the relevance of important terms identified by Sysrev for our search is given. Maximum relevance is given to early and late MCI, which is fortunately also of utmost concern in this work. A total of 215 articles were acquired, including 161 articles from Science Direct, 15 articles from the Web of Science, 6 articles from PubMed, 25 articles from ACM, and 8 articles from IEEE Xplore. The documents include all types of documents, such as research papers published in peer reviewed journals or conference proceedings, book chapters, survey papers, theses, patents, handbooks, or encyclopedias. The articles were further studied and analyzed according to the inclusion and exclusion criteria. A study was included upon the common agreement of all authors. Authors took extensive care to avoid redundancy during the filtering process. The screening process was supported by Sysrev [22].

### 2.3. Inclusion and Exclusion Criteria

The inclusion and exclusion criteria are critical to selecting the papers that are significant and aligned with the study objectives. We included articles that implemented DL architecture with TL and neuroimaging biomarkers for the diagnosis and early detection of AD. The issues in the conventional ML, DL and hybrid methods for the early detection of AD by using neuroimaging data, which enforced this inclusion criterion, are given below:The accuracy achieved by ML, DL and hybrid methods is summarized in Table 1. While traditional ML methods such as SVM have shown good results, DL methods such as CNN and sparse autoencoders have surpassed ML methods. Hybrid methods use deep learning for feature extraction and traditional ML methods such as SVM for classification. The combination of SVM for classification and stacked autoencoder for feature selection produced accuracies of up to 83.3% for early detection of AD. The maximum accuracy achieved by DL method for early detection of AD is 82.9. This has been further improved by using TL, as is evident in Section 3.One issue with traditional ML and DL models is that they utilize samples only from a single domain, which reduces their accuracy, when the numbers of samples is lower. TL utilizes samples from the target domain as well as from various supplementary domains.In traditional ML, results are influenced by well-defined features; selecting the optimal features from the highly dimensional neuroimaging data required the involvement of experts and optimization in multiple stages, which may be time consuming. DL identifies important features from neuroimaging data automatically, but the problem in training a deep neural network model from scratch is that it can take too long to converge and needs a dataset of sufficient size. Image classification datasets used for object classification have millions of images, but neuroimaging datasets are limited to hundreds. Specifically, the size of datasets for sMCI and pMCI subjects is very limited, and physician’s interpreted data can be expensive or inaccessible due to ethical reasons. A limited number of training data can lead to overfitting.The hybrid methods yield good performance with a limited amount of data, but they do not take complete advantage of DL. Here, the idea of TL by using pre-trained CNN networks came into the picture. Pre-trained CNN networks are trained on datasets of millions of images, on multiple servers for weeks, and facilitated by professional experts for public use. These networks can be used with small datasets by performing the fine tuning of the final layers of the CNN. Even TL has been implemented for cross-domain applications in the included studies.

The authors believe that the clinically acceptable system for the early detection of AD will make exhaustive use of the TL concept. It can be seen in the further discussion that the recent publications with good accuracy for the early detection of AD used TL.

Cross validation and at least two prediction matrices were also considered. The clear explanation of data sources and pre-processing methods applied for biomarkers were also considered as one of the inclusion criteria. Articles such as survey papers, thesis, patents, handbooks, encyclopedias were excluded. We did not find any book chapters or conference proceedings that satisfied the inclusion criteria.

### 2.4. Study Selection

The search strategy and study selection were delineated in detail using the PRISMA flow diagram [20] (Figure 2). After the identification of the articles through the search strategy, 215 articles were obtained. In total, 15 duplicates were removed. The remaining 200 items were screened for a second time. The studies’ importance was identified through the title, abstract, and 165 articles were excluded as they met at least one of the exclusion criteria. Several papers were excluded as they classified other diseases, such as autism and breast cancer, and did not focus on Alzheimer’s disease. Out of the 35 articles retrieved, in total,12 articles [36,37,38,39,40,41,42,43,44,45,46,47] were excluded as they did not use TL approaches or did not perform the prognostic prediction of AD. Five articles [48,49,50,51,52] were excluded as they did not provide at least two prediction matrices or did not perform cross-validation of the data. Two studies [53,54] were excluded as they did not provide the insights for input management. The other two papers [55,56] were excluded as they did not use neuroimaging data. Finally, the study proposed by [57] was excluded as they tried to distinguish AD from Parkinson’s. The remaining 13 papers [58,59,60,61,62,63,64,65,66,67,68,69,70] were included in this systematic review process. These studies are relevant to answering the research questions identified at the start of this section. In this research area, the leading journals are *Journal of Neuroscience Methods*, *NeuroImage: Clinical and Behavioural Brain Research*, and *Medical Image Analysis*. According to Google Scholar, references [58,65,66,68] are the most cited publications.

### 2.5. Data Extraction Parameters

The data collected were extracted based on the following parameters to conduct the systematic review.

Neuroimaging modality.DL algorithm and TL approach.Input management techniques.Pre-processing techniques for neuroimaging biomarkers.Dataset sources and number of neuroimaging biomarkers.Data validation and augmentation techniques.Software used for the pre-processing of neuroimaging biomarkers and implementing algorithmsClassifier type.Performance metrics of the proposed system.

### 2.6. Risk of Bias

The main limitation of conducting a systematic literature review is that it might be biased. To avoid this scenario to the maximum possible extent, the authors designed the search key using the keywords and logic operators. Based on the PRISMA guidelines, the authors tried to follow the best possible criteria and processes for completing the review. Only articles that satisfied all the inclusion criteria were included in the final evaluation. Although early researchers conducted several reviews [71,72] of ML and DL methods for AD classification, they did not consider all these RQs, specifically RQ1 andRQ4. In [71], the authors conducted an in-depth analysis of studies published between 2013 and 2018, and they performed a comparison of the diagnostic classification accuracy of pure DL and a combination of DL and ML methods. They also compared the change in accuracy based on the type of biomarkers. In [72], the authors presented the review of studies published between 2005 and 2019and conducted an in-depth analysis according to the three categories, namely state vector machine, artificial neural network, and deep learning models. The main differences in the existing studies and this work are that in the previous studies, the authors did not perform the comparative analysis of the change in accuracy according to the DL and TL combinations, and according to the type of biomarker and input management technique. Therefore, this study can help future researchers in identifying the most effective combination of the DL model with TL and in choosing the most suitable preprocessing method. Additionally, in the previous studies, no discussion was evident regarding the emerging biomarkers such as amyloid-PET and tau-PET. In the current study, the authors explored the importance of these biomarkers in future research for the early diagnosis of AD.

In Section 3, answers to all the RQs are discussed in detail. We hope that this work will provide useful information for the scientific community in understanding the research gaps, opportunities, and challenges in this field, which will help researchers in implementing a clinically acceptable solution for the early diagnostic of AD. Several relevant acronyms are provided in the Appendix A.

## 3. Results and Discussion

This systematic review includes 13 studies from four different databases. Among these, nine studies were included from Science Direct, two from IEEE Xplore, one from PubMed, and one from Web of Science. Based on the keywords, we found acceptable studies to have been published only from 2017 to 2020—one study in 2017, five studies in 2018, three studies in 2019, and four studies in 2020. The details regarding the included work, year, journal name, database, country, and citations are provided in the Appendix A. The review process, which focused on extracting information from the included studies, is presented in the Appendix A. Appendix A presents the technical insights and different datasets of neuroimaging biomarkers of the included studies. Appendix A provides a category-wise analysis of different pre-processing techniques for neuroimaging biomarkers with the required software.

### 3.1. Answer to RQ1

The first research question concerned the types of DL architectures and TL methods used to capture the disease-related patterns from neuroimaging modalities to classify AD and predict the conversion of MCI to AD. DL architectures can be divided into two categories, either supervised learning or unsupervised learning. The unsupervised methods can be divided into convolutional autoencoders (CAEs), stacked autoencoders (SAEs), and inception convolutional autoencoder (ICAEs). The supervised methods can be divided into DenseNets, deep neural networks (DNNs), recurrent neural networks (RNNs), Google Net, CaffeNet, AlexNet, and 2D/3D convolutional neural networks. In Figure 3, the dissemination of different DL architectures and their combinations used in the analyzed studies is given.

#### 3.1.1. Unsupervised Deep Learning

Unsupervised learning is an indispensable component when confronted with data shortages and high dimensionality. It is used to attain a task-specific depiction from neuroimaging data and for transfer learning, particularly beneficial when one has restricted labeled data but a bigger set of unlabeled data. Autoencoders (AEs are the most common approach to be used in such cases. In general, numerous AEs are piled on top of each other, and the result is called a stacked AE. Researchers have used different variations of AE, such as CAE, ICAE, and SAE, for feature extraction and dimension reduction, and transfer learning in an unsupervised way. KanghanOh et al. [60] proposed in their work CAE- and ICAE-centered unsupervised learning to extract sparse representations for AD and NC. Initially, MRI scans of patients with AD and NCs were pre-trained based on CAE-based unsupervised learning, followed by supervised fine-tuning to build the classifier to differentiate AD vs. NC. After that, they transfer learned knowledge for classifying the pMCI vs. sMCI.

In [60], the authors applied an inception module to the convolutional autoencoder (ICAE) to discover highly advanced representations and lessen the dependency on the fully connected layer. The outcomes of the CAE and ICAE-based models exhibited improved performances, as compared to the baseline models [73,74,75]. Donghuan Lu et al. [65] used SAE for the pre-training of each DNN of the proposed multiscale DNN. They used greedy layer-wise training [76], training a single hidden layer at a time. This was followed by supervised fine-tuning that was steered to build the classifier to differentiate pMCI vs. sMCI, which showed better performance than the published literature.

#### 3.1.2. Supervised Deep Learning

Researchers have used the combination of unsupervised learning for the future extraction and initialization of weight and biases of the target networks, with supervised fine-tuning as explained below.

Deep Neural Networks: DNNs are the most extensively used in diverse research areas to learn formerly unknown abstract patterns. DNN provide some of the top performances amongst the machine learning methods recommended for recognition tasks. Deep learning networks have also been applied to distinguish the AD-related progression patterns. Liu et al. [77] trained a stacked autoencoder to study the concealed representation trailed by a softmax output layer for classification. Donghuan Lu et al. [65] suggested a novel framework comprising of multiscale DNN to study the patterns of metabolism alterations due to AD pathology as discriminative from the patterns of metabolism in normal controls. For a single DNN, they implemented training in two steps—firstly, unsupervised pre-training by using SAE, and secondly, supervised fine-tuning by keeping the three encoder layers and adding a softmax output layer for each DNN. For improving the accuracy of the progression prediction of AD, the authors used the weights of the DNN, used to classify AD vs. NC.

Convolutional Neural Networks: CNNs are stimulated by the visual cortex of the brain. CNNs are the most effective model for image analysis. They are used to better exploit spatial information by taking 2D/3D images as the input and extracting features by stacking convolutional layers. The main advantage of a CNN is to amalgamate feature extraction and classification. In comparison to DNNs, the number of parameters is drastically reduced because of shared weights and pooling layers in CNNs. The necessity for a large dataset can be contemplated as a flaw of these models. The CNN’s architecture received increased attention after a deep CNN achieved excellent results in the ILSVRC competition. Although many studies preferred to construct their own CNN structures, it is common in the literature to use well-established pre-trained CNN structures such as ResNet, deep-ResNet [78], CaffeNet [79], AlexNet [80], DenseNet [81], VGG16 [82], GoogleNet [83], and inception-v4 [84] for transfer learning. CNNs are designed to identify patterns in 2D images. Several researchers have used 2D CNNs for 3D neuroimaging data. Farheen Ramzan et al. [59] use ResNet-18 for transfer learning followed by a 2D CNN for the classification of AD vs. NC classes. They executed transfer learning in two approaches. Firstly, TL is used by substituting the last dense layer of the original network with the new dense layer to counterpart the number of classes. Secondly, all the layers, excluding the last layer (classifier) of the network, are used for feature extraction, and the weights of the last layer are reformed to the new task. Moreover, fine-tuning, in which more than one layer of the network is re-trained from the samples of the new task. Lan Lin et al. [64] extracted intermediate representations of structural MRI (sMRI) by means of a pre-trained 2D CNN model (CaffeNet). Moreover, they combined principal component analysis (PCA) and sequential feature selection (SFS). Finally, they adopted a support vector machine (SVM) for the progression prediction of MCI to AD. As neuroimaging data are 3D, and there is a spatial relationship amongst the images, 3D CNNs were used in the most recent research. For AD detection, CNNs essentially take the whole image or region of interest as the input. This may necessitate training an enormous number of parameters on a small dataset, which may lead to overfitting. To avoid overfitting, researchers used augmentation and cross-validation techniques [58,59,60,61,63,69]. Silvia Basaia et al. [59] used a 3D CNN without any previous feature engineering and irrespective of the erraticism of imaging protocols and scanners. To improve the performance of the prognostic progression of AD, they transferred the weights of the CNN used to classify AD vs. NC to the other CNNs to classify sMCI vs.pMCI. Their network architecture contains 12 recurrent blocks of convolutional layers (2 blocks with 50 kernels of size 5 × 5 × 5 with alternating strides 1 and 2 and 10 blocks with 100 to 1600 kernels of size 3 × 3 × 3 with alternating strides 1 and 2), a rectified linear unit (activation layer), a fully connected layer and one output (logistic regression) layer. Anees Abrol et al. [62] proposed an adapted form of ResNet for studying neuroimaging data to predict MCI to AD progression. The proposed method starts by training the deep models using MCI individuals only, followed by a TL version that is further trained on AD and controls. This framework has 3D convolutional units, 3D batch-normalization units, and non-linear activation units, and a fully connected (FC) layer featuring 512 output nodes. Marcia Hon et al. [68] used two different pre-trained 3D CNN-based architectures VGG16 and inception V4. Only the last fully connected layer is re-trained with the training data. For the transfer learning models, pre-trained network weights from ImageNet were obtained. FeiGao et.al [70] proposed an age-adjusted 3D CNN-based pre-training model. The proposed method aims to extract and transfer features. Moreover, the authors used a pre-trained network for feature extraction and age prediction and a fine-tuned network to transfer both features and knowledge from age prediction for MCI converter prediction. The pre-trained network takes 3D MRI images from NC subjects as inputs and predicts age and extracts related features. In the fine-tuning model, the age-related information from the pre-training is transferred. Together with the features from the pre-training model, the risk of the MCI subject converting to AD is predicted. Junhao Wen et al. [61] implemented four different CNN architectures—3D subject-level CNN,3D ROI-based CNN,3D patch-level CNN, and 2D slice-level CNN. Moreover, two different approaches were used for transfer learning: (i) AE pre-training for 3D CNNs, and(ii) ResNet pre-trained on ImageNet for 2D CNNs. Three approaches (3D subject-level, 3D ROI-based, 3D patch-level) provided approximately the same level of performance. On the other hand, the 2D-slice approach was less efficient.

Zhiguang Yang et al. [67] used the AlexNet pre-trained CNN architecture of the CAFFE framework [85]. CAFFE was implemented in C++ language and had Python and Matlab interfaces, and it could speedily realize the CNN on GPU and expediently convert between GPU and CPU. The three basic results of the CAFFE framework are blobs, layers, and nets. The network structure of AlexNet included eight layers of neural networks, five convolution layers, and three fully connected layers. Congling Wu et al. [63] explored and evaluated two CNN architectures, GoogleNet and CaffeNet, in multiple classifications and estimations of conversion risk through TL from pre-trained ImageNet (via fine-tuning). CaffeNet is a revised version of the classic CNN architecture of AlexNet [79]. GoogleNet model is the current state-of-the-art CNN framework proposed for the ILSVRC challenge. The architecture of GoogleNet [83] is considerably composite and deep, involving a 22-layer architecture. CaffeNet attained promising accuracies in the NC, sMCI, and pMCI classifications in comparison to GoogleNet.

Recurrent Neural Networks: RNNs [86] can efficiently model the temporal behavior for a time sequence, with the output being reliant on the preceding computations. Fan Li et al. [50] proposed a hybrid method including convolutional and recurrent neural networks for hippocampus analysis. They proposed 3D DenseNets based on the decomposed patches of the 3D image of the hippocampus to assess the intensity and shaping of the features and applied the bidirectional GRU (BGRU), which consists of a forward GRU and a backward GRU, to capture correlation features between the left and right hippocampus from two directions. The BGRUs are cascaded on the DenseNet and the inputs of each BGRU are the features of the left and right hippocampus learned by 3D DenseNets from both the interior and exterior hippocampus.

#### 3.1.3. Transfer Learning Methods

Training a DL model from the initial stage was performed in some of the studies. However, the training process consumes considerable time, and the size of neuroimaging datasets is restricted, which may lead to overfitting. As we have seen in the previous discussion of DL architectures, TL is less time-consuming and shows better performance in comparison to training from scratch. Recent studies [58,62,63,64,67,68,69] used initialization or feature extraction by CNN-based TL architectures, such as ResNet, deep ResNet, CaffeNet, AlexNet, DenseNets, VGG16, and inception with or without fine-tuning, followed by an extended network architecture. On the other hand, other approaches [59,65,66,70] used local TL. The weights of the CNN used to classify AD vs. NC were transferred to the other CNNs and used as initial weights to predict the progression of AD. Furthermore, Refs [60,61] used unsupervised DL architectures such as SAE and ICAE for the pre-training of 3D CNNs. Figure 4 shows the distribution of each TL approach and their combinations used in the analyzed studies.

#### 3.1.4. Comparative Analysis of Supervised/Unsupervised DL Networks with TL

According to our review, the unsupervised DL architectures SAE, CAE, and ICAE are best for extracting features and for classification. They can characterize highly non-linear and complex patterns and can be used for finding initialization parameters for CNNs. AEs have limitations, as they learn to capture as much information as possible and do not only capture relevant information.

Supervised methods were mostly used in the analyzed studies, as they enabled feature selection and categorization being combined into a single algorithm.TL methods have been used in the most suitable initialization and most informative feature extraction methods, followed by extended supervised DL architectures. 3D CNNs can obtain 3D neuroimaging information and has exhibited better performance in local feature extraction when used with ResNet for TL. However, the intricacy of training is a limitation, which can be solved by using ROI or patch-based pre-processing methods for neuroimaging modalities. On the contrary, 2D CNN is easier to train, but is not competent enough in encoding the spatial information of 3D images across the third dimension. Therefore, several studies use all three views of neuroimaging images or use RNNs to capture 3D information.RNNs show reliable performance for sequential 2D image slices. However, they have problems associated with the training process owing to the fading gradients. DNNs show a decent performance for vector-based problems, as they detect complex non-linear relationships, but have a sluggish training process, generalization issues, and are not ideal for images. The performance of the different DL architectures for the included studies concerning the accuracy, sensitivity, and specificity, and technical insights related to DL architectures and TL methods are reported in the Appendix A. The accuracy achieved to date by using different combinations of DL and TL approaches is discussed in Section 3.4.

### 3.2. Answer to RQ2

The second research question concerned the neuroimaging biomarkers and parameters used for the classification and prediction of the progression of AD. For AD detection, non-invasive biomarkers, such as MRI and PET, have been used commonly. During the feature selection process, the most commonly chosen features from these biomarkers are the mean sub cortical volumes, gray matter densities, cortical thickness, brain glucose metabolism, and cerebral amyloid-β accumulation in regions of interest. The neuroimaging data acquired from these technologies have been used for providing a computer-aided system to help physicians to improve healthcare systems. MRI is well-known and uses biomarkers and has demonstrated excellent performance in the implementations of AD vs. NC classifications. MRI provides the ability to study pathological brain alterations accompanying AD. It uses a magnetic field and radiofrequency pulses to generate a 3D representation of soft tissues, bones. Different types of MRI images have been used in the included studies. The sMRI was used in [62,63,64,68]. sMRI facilitates the tracing of brain atrophy and aids in finding out the possible cause of AD; it has additional advantages related to its high spatial resolution and its availability. Functional MRI (fMRI) also reflects the changes associated with blood flow. Resting-state fMRI (rs-fMRI) captures the fluctuations in blood oxygenation levels of subjects in the rest state. As a result, the brain regions affected by neuron-degeneration demonstrate diverse patterns of blood oxygenation levels. The fMRI has been used in one included study [58]. Several of the authors have used the T1-weighted sequence(T1w) of sMRI scans. T1w sequences are part of the MRI protocols and are thought of as the most anatomical of images. These sequences approximate the closest appearance of tissues such as white matter (WM), gray matter (GM), and cerebrospinal fluid (CSF) [87]. These measurements are one of the most significant biomarkers to detect AD progression [68]. T1w images were used in [59,60,61,69,70]. The T1w sequences used are of different types, for example, 3T or 1.5 T baseline. Here, 3T/1.5T implies a 3Tesla/1.5 Tesla MRI, which is produced by a magnetic field of 3 Tesla or 1.5 Tesla and results in a cleaner and more complete image [88]. The authors of [89] used T1w MP-RAGE sequences of MRI and obtained excellent results. Magnetization-prepared rapid gradient echo (MP-RAGE) is a sequence of sMRI which captures high tissue contrast and offers high spatial resolution with whole-brain coverage in a short time period. PET is nuclear medicine functional imaging technology used to observe the metabolic process for AD detection. It has been used for many years for research and clinical purposes. Meanwhile, 18F-FDG-PET is the most widely used type of PET image. It provides a measure of cellular glucose metabolism. It can be used to assist the neurocognitive disorder due to AD [90]. It is mainly beneficial for the early detection of AD, as it can demonstrate the distinctive patterns of AD earlier them MRI for MCI subjects [91]. Another type of PET is amyloid-PET, which is used to assess brain amyloid deposition, one of the main neuropathological milestones of AD, with an elevated sensitivity and specificity in patients with established Alzheimer’s disease who underwent an autopsy within one year of PET imaging [91]. Several studies used different combinations of biomarkers to improve the accuracy of AD detection [92,93].PET images were used in [65,66,67]. In [65], the authors used18F-FDG-PET with sMRI, and in [66], the combination of 18F-FDG-PET with AV-45 PET obtained comparatively good accuracy for detecting the progression of AD. Another type of future PET is tau-PET imaging [94]. Abnormal deposition of tau is a key contributing factor to neurodegenerative diseases such as AD. The advances in neuroimaging technologies have steered the advancement of tau-specific tracers for PET such as THK5351 and THK5357 [95]. These images are specifically helpful in monitoring the progression of the disease. We did not find any article where the authors used tau-PET for predicting the AD progression by using DL and TL methods. Additionally, a few additional factors that are pertinent to AD detection are gender, age, the pattern of speech, EEG, tau protein, Aβ protein, retinal abnormalities, postural kinematics analysis, MMSE and CDR score, logical memory test, and some genes believed to be responsible for AD [23]. A detailed explanation of the discussed neuroimaging modalities for AD detection was given in [91]. Figure 5 presents healthy and AD images. Their clinical acceptability decreases from left to right. In the sMRI, grey matter volume is displayed in blue, in the 18F-FDG PET images, reduced metabolism is displayed in green, in the amyloid-PET images, the small quantity of deposited amyloid is displayed in green or blue, and in the tau-PET images of AD patients, low tau-tracer retention is displayed in green or blue. Two other types of future PET images are TSPO-PET [95] and SV2A-PET [96], which can be integrated into future research with DL techniques for the detection of AD progression.

The distribution of neuroimaging modalities used in the included studies is given in Table 2.

In Table 3, the accuracies achieved by the type of neuroimaging biomarkers in the included studies are given.

### 3.3. Answer to RQ3

The third research question focuses on the used pre-processing techniques and the input management of neuroimaging modalities for the classification and prediction of the progression of AD. Pre-processing is one of the main contributing factors to determine the success of any AI-based system. It removes artifacts and noise from the data to enhance the quality of the image and improves feature extraction. Pre-processing steps are not needed in some DL architectures [98,99]. Nevertheless, most of the included studies used some of the following pre-processing methods on the raw neuroimaging data.

1. Intensity Normalization: The use of different scanners and parameters for examining subjects at different times may cause large variations in the intensities. It can cause performance degradation in subsequent processing, such as registration, segmentation, and tissue volume measurement [99]. Normalization relates to plotting the intensities of all the voxels against a reference scale so that alike structures have identical intensities. The most common method is to use the non-parametric, non-uniform intensity, normalization (N3) algorithm. This is used for correcting the intensity’s non-uniformity, by sharpening histogram peaks [100]. It has been applied in the included articles [58,63,69]. In [60], the authors normalized the intensities of the MRI scan to [0, 1]. In [66], the brainstem region was utilized for internal normalization, as it is expected not to be affected by AD, hence the mean intensity in the brainstem region is calculated and used to distribute the average intensities respectively for each subject.

2. Spatial Smoothing: An FWHM Gaussian filter between 5 and 8 mm reduces noise level and preserves the signals present in the image [101]. It was applied in three of the included studies [58,62,64].

3. Registration (Spatial Normalization): Registration is the process of mapping a subject’s neuroimaging image to a reference brain space. It allows comparison among subjects of varying anatomy [102]. It standardizes neuroimaging images regarding a common template such as MNI [103]. The authors registered the neuroimaging images into the MNI template by using different methods. In [58], linear transformation with 12 DOF (such as translation, scaling, shear, and rotation) was used. The Diffeomorphic Anatomical Registration Exponentiated Lie Algebra (DARTEL) registration [104] was used in several studies [59,60,64]. In [62], linear (affine) registration was performed using the SyN algorithm [105]. In [61], the authors used non-linear registration by using the unified segmentation approach [106]. In [62], the images were spatially normalized to the 152 average T1 MNI template. In [65], the authors performed image registration via a highly dimensional, highly accurate non-rigid registration method (LDDMM) [107], widely thought of as being the prominent registration tool. Another registration process found in the literature is co-registering multiple modalities. Anterior commissure (AC) and posterior commissure (PC) are two main landmarks in the brain. AC-PC line [108] has been embraced as a standard by the neuroimaging community, and in many examples, the reference plane for axial imaging to achieve the comparison between different subjects. In [66], the authors registered PET images such that the forward-facing and the rearmost axis of the subjects were parallel to the AC–PC line. In [69], the authors implemented the affine registration method [109] to linearly align the MR images to a template. In [70], the authors conducted rigid registration to the MNI152 space to ensure consistency of orientation and position. The authors of [64,67,70] applied interpolation methods [110] to transfer data into the same voxel sizes and dimensions.

4. Gradwrap: Correct geometry distortion due to gradient non-linearity. It was used in one included study [63].

5. Tissue segmentation: Tissue segmentation aims at partitioning a neuroimaging modality into segments corresponding to different tissues (grey matter (GM), white matter (WM), and cerebrospinal fluid (CSF)), probability maps and metabolic intensities of the brain region in the case of PET images. Probability maps are used as the input of the classification model. Probability maps contribute the quantitative measurement of the spatial distribution of these tissues in the brain. Neurodegeneration causes changes in the volume of GM at its initial stages, specifically in the temporal lobe area [111]. This process was conducted in [59,62,64,65,67] using different software tools.

6. Bias-field correction: This is a low-frequency, smooth signal that degrades neuroimaging images, particularly those formed by old MRI machines [112]. A pre-processing step is needed for correcting bias-field inhomogeneities. This process was applied in three of the included studies [60,61,63] by using the N41TK [113] algorithm.

7. BrainExtraction (Skull Stripping): For eliminating non-brain tissues, such as neck and skull tissues. It was used in three of the included studies [58,61,69]. In [61], the authors applied a mask-based approach for skull-striping.

8. Motion Correction: Used to eliminate and make the impact of head motions more precise during the data acquisition process. It was used in one included study [58].

9. High-pass filtering: Used to eliminate the low-frequency noise signal, which is generated because of psychological artifacts. It was used in one of the included studies [58].

In [68], the authors used the entropy-based sorting mechanism to choose the most informative images from the axial plane of each 3D MRI scan, and they did not use any other pre-processing methods.

In Table 4, the dissemination of each pre-processing method from the included studies is given; as we can see, almost 50% of studies performed intensity normalization and registration.

Based on the extracted features from the neuroimaging modalities, the management of input data can be allocated to the following four categories.

1. Voxel-based input management: In this approach, voxel intensity values of the complete neuroimaging modalities or tissue components are used. The registration method is necessary for this approach to map all the images into a standard 3D space. In [59], the authors performed tissue segmentation and registration on T1-weighted MRI to generate the GM, WM, and CSF tissue probability maps in MNI space. Several research studies [60,61,70] performed a full brain image analysis of T1W sMRI. In [66], the authors performed the full brain analysis in multimodality mode (18 FDG-PET and AV-45 PET) and co-registration by using the AC–PC line. The advantage of full brain analysis is that the spatial information is fully integrated, and we can obtain 3D information from neuroimaging scans. The drawback of this approach is the high dimensionality and computational load. In [64], the authors segmented the sMRI images into GM, WM, and CSF by using the DARTEL method and used PCA and SFS for the precise selection of the best features.

2. Slice-based input management: In this approach, 2D slices are extracted from 3D brain scans either by using the researcher’s logic or standard projections such as the horizontal plane, frontal plane, and median plane. Full brain analysis is not possible, as all the information of 3D brain scans cannot be converted into a 2D slice. For tissue segmentation, generally, slice-based methods take tissues from the central part of the brain. In [58], the authors performed many pre-processing steps, as given in Appendix A, to convert 3D rs-fMRI scans to 2D images, along with the image height and time axis, and they obtained 6160 2D images for each fMRI scan and saved them in PNG format. In [63], the authors used Matlab and NifTi to convert 3DMRI scans of each subject into 160 2D image slices. In [67], the authors converted the NifTi images (created from the gray matter density maps of each subject) averaged from the Z-axis into 65 PNG images. In [68], the authors performed the entropy-based sorting mechanism to select the 32 most informative images from the axial plane view of each 3D scan. The advantage of this approach is to decrease the number of parameters to thousands from millions during the training and testing phase of the model. Because of using 2D representations of 3D scans, the adjacent 2D images lose spatial dependency.

3. ROI-based input management: Studies based on the region of interest focused on the areas that are identified to be informative. In this context, the informative parts that are affected in the early stage of AD have to be recognized. The hippocampus, amygdala, and entorhinal cortex, grey matter in the entire brain, and the temporal and parietal lobe must be treated with greater precedence for AD or MCI classification, whereas the amygdala and hippocampus can be used for predicting the conversion of MCI to AD [114]. Meanwhile, the occipital lobe, thalamus, globus pallidus, and putamen ought to be low priority choices for the early diagnosis of AD. This approach also requires the previous knowledge of the brain atlas, such as automated anatomical labeling (AAL) [115] or Kabbani reference work [116]. In [62], the authors identified the utmost discriminative brain regions for classifying the pMCI vs. sMCI by approximating occlusion sensitivity utilizing the network occlusion approach [117]. They occluded brain networks in reference to the AAL brain atlas one at a time, and the significance of each brain region was assessed. The most relevant weights were perceived in the “hippocampus, temporal superior, par hippocampal gyrus, middle and inferior gyrus, occipital superior, fusiform gyrus, middle and inferior gyrus including cuneus and calcarine, lingual gyrus, frontal middle, and inferior gyrus regions, precuneus, and cerebellum 6, crux 1 and 2 regions” [62]. They concluded that the hippocampus and amygdala subcortical regions in the medial temporal lobe are the most informative regions in the early AD. In [65], the authors segmented the sMRI into GM and WM and divided the GM into 85 cortical and subcortical anatomical ROIs. In [69], the authors performed the segmentation of the hippocampus from other regions and create a binary mask for the individual hippocampus. After that, the centroid of each hippocampus was calculated, and patches were extracted from the centroid for further implementation. The ROI-based approach has the advantage of low feature dimension and easy interpretation, but it ignores all the details of abnormalities. Still, research is ongoing to identify the most informative or affected part of the brain due to AD [33,118].

4. Patch-Based Input Management: A patch is a 3D cube; in this approach, disease-related patterns are identified by mining the features from patches. The core issue is to select the most discriminative patches for expressing both patch-level and image-level features [32]. In [65], the segmented ROI is additionally sectioned into smaller regions of variable dimensions, termed as patches, for implementing the multiscale method. This method states that the signal extracted at multiple scales from the fine-scale to the coarse-scale can be applied concurrently to detect deviations due to AD. The sizes of the patches were decided based on the consideration to retain sufficient comprehensive information as well as to avoid large feature dimensions. The subdivision of ROI into patches is performed by the k-means clustering algorithm [119], where ROI can be clustered by using spatial coordinates. In [69], a binary mask was created by means of the segmentation of each hippocampus, and 3D patches were created by cropping at the centroid of each mask. Three-dimensional patches decomposed further into two patches, for the internal and external hippocampus. The benefit of this approach is that it is sensitive to small changes, but selecting the most informative patch is still a challenge. The input management method-wise analysis of different pre-processing techniques for neuroimaging biomarkers has been summarized in Appendix A.

We conclude that input management is also a critical issue. A 2D-slice input has the advantage of fewer training parameters and a less complex network, but also has the drawback of losing spatial dependency between nearby slices. Voxel-based input management considers all the brain information but treats all regions equally, without any consideration of the anatomical structure, and has the drawback of high feature dimensionality and load on the processors. An ROI-based input is easily interpretable and has been used in clinical diagnosis, but in this method, the entire brain is represented by fewer features, and it provides knowledge about only a particular part of the brain, such as the hippocampus. Research is still ongoing to recognize the uses of other brain regions in detecting AD. Sometimes the ignorance of small abnormalities in the ROI may be the cause of damage to discriminative information and restrict the true command of extracted features. Nevertheless, patch-based input management can competently handle large feature dimensions and are responsive to small changes [120]. In Table 5, the distribution of input management approaches in the included studies is given.

### 3.4. Answer to RQ4

Out of 13, two studies performed multiclass classification and 11 studies performed binary classification. The accuracy performance metric was used consistently in all the included studies. In [58], the authors performed multiclass (AD, NC, sMCI, pMCI, SMC, MCI, respectively) classification by using the softmax classifier and rs-fMRI, and achieved an average accuracy of 97.92% and 97.98% for the off the shelf and fine-tuned models, respectively. In [63], the authors performed three-way discrimination among the NC, sMCI, and pMCI, respectively, by using the GoogleNet/softmax and CaffeNet/softmax, and obtained an overall accuracy score of 83.23% and 87.78%, respectively. In [59,60], the authors implemented all possible binary classifications for distinguishing NC, sMCI, and pMCI (AD vs. NC, NC vs. sMCI, pMCI vs. sMCI, NC vs. pMCI, and AD vs. sMCI, AD vs. pMCI) by usingT1wMRI. In [61], the authors performed binary classification for AD vs. NC and sMCI vs. pMCI by using three input management approaches (ROI-based, patch-based, subject-level); for implementing all these methods, three different datasets and T1wMRI were used. Datasets are discussed in detail in Section 3.5.In [62], the authors performed binary classifications tasks (AD vs. NC, sMCI vs. pMCI, NC vs. pMCI, AD vs. sMCI) by using ResNet, SVM classifier and sMRI. In [64], the authors performed only one binary classification task (sMCI vs.pMCI) by using CaffeNet and sMRI biomarkers. They combined PCA and SFS for feature selection and finally adopted SVM for the prediction. In [65], the authors used multimodality (FDG-PET metabolism imaging and sMRI) and an ensemble of classifiers, multiple networks were trained, and then they chose the final classification result (PCA followed by SVM) to identify subjects classes (AD vs. NC, sMCI vs. pMCI). In [66], the authors performed the binary classification for AD vs. NC and sMCI vs. pMCI by using multimodality(FDG and AV-45 PET) and softmax classifier. In [67], the authors performed the binary classification sMCI vs. pMCI by using FDG-PET images, AlexNet, and SFS feature selection followed by an SVM classifier. In [68], the authors used three different models (VGG16-from scratch, VGG16-TL, InceptionV4-TL), softmax classifier, and sMRI biomarkers for the binary classification of AD vs. NC. In [69], the authors conducted the multi-class classification (AD vs. NC, sMCI vs. pMCI, and sMCI vs. NC) by constructing DenseNets on the decomposed patches of the internal and external hippocampus of T1wMRI images, cascaded by RNN to learn the features from both hippocampi and the softmax classifier. The authors also compared the proposed method by implementing other commonly used hippocampus analysis methods such as disease detection by shape and volume analysis of the hippocampus. In [70], the authors performed a binary classification task (sMCI vs. pMCI) by extending feature-based transfer learning with knowledge transfer. They introduced a surrogate age marker that was captured through pre-training, and then they applied fine-tuning for the final prediction by combining the output of 3D CNN and surrogate age marker. They used T1wMRI and a fully connected layer obtained through flattening for predicting the value of surrogate age marker.

In Table 6, the accuracy achieved according to the DL and TL combinations is presented. It can be seen that the maximum accuracy achieved for classifying the sMCI vs.pMCI is 87.78%, obtained through the combination of CaffeNet (training) and ImageNet(transfer learning).

In Table 7, the accuracy achieved according to the input management and neuroimaging biomarkers is presented. It can be seen that the maximum accuracy achieved for classifying the sMCI vs. pMCI is 87.78%, obtained through the combination of slice-based and sMRI.

### 3.5. Answer to RQ5

Numerous online datasets of neuroimaging biomarkers are available that have been developed with the motive of providing the free availability of neuroimaging data to the community of scientists to assist in forthcoming discoveries in the neurosciences. Datasets such as ADNI [121], OASIS [122], AIBL [123], IXI [124], and MIRIAD [125] are freely available online.

ADNI: ADNI was developed by longitudinal multicenter studies. It was released in 2004 by the National Institute of Aging (NIA), the National Institute of Biomedical Imaging and Bioengineering (NIBIB), and some other national institutes as a USD 60 million, 5-year partnership project, based in North America. The acquisition of all the data was achieved through the ADNI protocol. To date, they have four variants: ADNI-1 (400 MCI,200 sMCI, and 200 NC contribute to the research and were followed for 2–3 years, with MRI and FDG-PET biomarkers only), ADNI-Grand Opportunities (the existing ADNI-1 cohort + 200 EMCI subjects), ADNI-2 (ADNI-1, ADNI-GO cohort + 150 NC,100 EMCI,150 LMCI, and 150 mild AD subjects,107 SMC subjects and also amyloid PET images have been added), and ADNI-3(which began in 2016 and will remain up to 2022, and tau-PET has already been added as a key indicator for AD, while further research is ongoing for the discovery, optimization, standardization, and validation of clinical trial measures for AD detection). In this work, this is the most used dataset, being used in 92% of studies by itself or in combination with other datasets.

OASIS: OASIS-3 is the latest release in the OASIS series. The formerly released OASIS-Cross-sectional and OASIS-Longitudinal datasets have been utilized for AD research, OASIS-3 is the longitudinal, cognitive biomarker dataset for normal aging and AD. It contains the neuroimaging data generated from 2168 MRI sessions and 1608 PET sessions by using 1098 subjects (609 NC and 489 people at different stages of cognitive debility) in the age range of 43-95 years. It was used in 16% of the studies in our review.

AIBL: This is known as Australian ADNI, and adds scientific value to the ADNI cohort. It contains 250 MR/flutemetamol, 200 MR/florbetapir, and 50 MR/PIB images. It was used in 8% of the studies in our review.

IXI: This contains 600 images from normal, healthy subjects. Researchers can download T1wMRI, T2wMRI, and DTI images directly in NIFTI format. It was utilized in 8% of the studies in our review.

MIRIAD: This is a database consisting of MRI scans of 46 AD and 23 NC subjects, and scans were collected at interims from 2 weeks to 2 years. This study has the objective to examine the viability of using MRI for clinical trials of AD.

Meanwhile, several studies preferred to use their own datasets. In [59], the authors designed a self-regulating dataset “MILAN” of 3D T1 weighted images, acquired from 229 subjects (124 AD,50 MCI,55 NC), recruited at the Neurology Department, Scientific Institute, and University Vita-Salute San Raffaele, Milan. The distribution of the discussed datasets in the included studies is given in Table 8.

Only some studies used the combination of different types of datasets. For developing a clinically acceptable system, researchers should use data of different homogeneities [126]. The size of the datasets must be balanced either by taking the same number of images for training and testing or by redefining the loss function. Other points such as patient overlapping, set sampling or ground truth should also be considered for the uniform distribution of data and avoiding data leakage during training and testing. Techniques for avoiding data leakage and overfitting are discussed in Section 3.7. Furthermore, all the other details, such as sample size and the combination of different datasets from all included articles, has been summarized in Appendix A.

### 3.6. Answer to RQ 6

Different software packages are available to assist researchers in perfoming the pre-processing of neuroimaging data and the implementation of DL and TL methods. The following brain image analysis packages were used for the pre-processing of neuroimaging biomarkers in the included articles. The pre-processing pipeline implemented by these packages is given in Appendix A.

FSL [127]: This is the all-inclusive library of tools for analyzing sMRI, fMRI, and DTI neuroimaging data. Most of the tools can be run either by command line or GUI interface. It was used in [58,59].

SPM12 [128]: Thus is intended for the analysis of neuroimaging data sequences; the sequences can be a succession of images from diverse allies. The current release, updated on 13 January 2020, is specifically intended for the analysis of fMRI and PET images. It was used in [59,60,62,64].

Nipype [129]: This is an open-source Python project developed by the Nipy community [130], provides an even interface to the currently available brain imaging software, and facilitates communication amongst these packages (for example, AFNI [131], ANTS [132], Camino [133], FreeSurfer [134], FSL, MNE [135], Slicer [136], SPM [137]) within a single flow. It eases the learning curve required to use diverse packages. One of the most important advantages is that it makes one’s research reproducible and shares one’s processing workflows with the research community. It was used in [61].

MATLAB [138]: The image processing toolbox permits one’s to mechanize common image processing workflows. The researcher can section image data, perform batch processing for large datasets, perform the registration of images, create histograms, and manipulate ROIs. MATLAB was used in [63,68].

NifTi_2014 toolkit [139]: This is a tool for analyzing and processing neuroimaging images that can be loaded by MRIcro software [140] and MATLAB2015b [141]. It was used in [67].

FreeSurfer [134]: This is an open-source collection for analyzing and processing MRI images, developed by the Laboratory for Computational Neuroimaging (LCN), USA. It is a set of automated tools for the renewal of the brain cortical surface from sMRI and the overlap of fMRI onto the recreated surfaces. It was used in [65].

MRIcron [140]: This allows users to view medical images in various formats, and creates format headers for exporting neuroimaging biomarkers to other platforms. It is a stand-alone program but includes tools to analyze MRI, fMRI, and PET images. It can be used for the efficient viewing and exporting of neuroimaging data and to identify ROIs. It was used in [70].

Moreover, in [66], the authors did not perform any pre-processing of FDG and AV-45 PET images. Pre-processed images were downloaded from the ADNI at the most progressive pre-processing stage, and were used for deep CNN training and testing. The dissemination of pre-processing software in the included articles is given in Table 9.

Software packages such as CAFFE [85], Keras [142], Tensorflow [143], Theano [144], Pytorch [145], MatConvNet [146], and the Deep Learning toolbox [147] have been used for implementing DL and TL algorithms. In Table 10, details of the packages used in the included articles are given. The distribution of software packages in the included articles is given in Table 11.

### 3.7. Answer to RQ7

Dataset size has a notable impact on the performance of classifiers on an unseen test dataset [44]. The available datasets of AD and MCI subjects are of a relatively small size, with only a few hundred samples. DL algorithms tend to be easily over-fitted when trained on a lower number of samples due to a large number of training parameters. In this section, we discuss the techniques used in the included articles for reducing overfitting.

Data Augmentation: This is a way of increasing the heterogeneity of training datasets without collecting new data. It generates new data samples from the existing data. It can be categorized into two methods. The first category is transformation methods, which cover a mixture of simple transformations such as random translation, rotation, reflection, distortion, blurring and flipping, cropping, noise injection, gamma correction, scaling, and intensity variations by arbitrarily adjusting brightness, contrast, saturation, and hue on the training data. Transformation methods were used in [59,60] to improve the classifier performance. The second type of method is neuroimaging data synthesis [148], intended to generate a new dataset that shares features with the source dataset [68]. It can be implemented by using autoencoders (AE) [149] and generative adversarial networks (GAN) [150,151] for neuroimaging biomarkers. Nevertheless, this area needs to be explored and the effectiveness of the synthesis images for predicting AD still must be proved. In [58], the authors obtained 6120 2D images from each fMRI scan by using pre-processing techniques. In [63], the authors employed a novel strategy for data augmentation, based on the image integration method Shin [152], to create sample image patches from MRI. It uses the information from training data adequately to implement the constraint of the three-channel input of CNN. In [66], the authors performed the augmentation of PET images by flicking them in the left–right direction. In [69], the hippocampus centroid is shifted by ±2 voxels in the x and z directions to extract more patches. Technical insights of the augmentation approaches from included articles are given in Appendix A. Nevertheless, some studies did not perform augmentation at all [62,64,65,67,68,70], and relied only on TL methods to reduce overfitting.

Transfer Learning: This is the idea of utilizing a model trained on a certain chore (for example, the ImageNet classification task or unsupervised learning task), conducive to achieving the task of AD classification or predicting the progression of MCI to AD. TL methods may reduce the overfitting by providing a better weight initialization of DL models. It also reduces the time needed for training the DL models and provides better performance compared to training from scrap [153].

Regularization: This can be performed by using different approaches, such as dropout, weight decay, and L1/L2 regularization. Dropout relies on the idea of arbitrarily and independently dropping neurons, setting their output value equal to zero, which makes the network less intricate and less vulnerable to overfitting. It improves the generalization capability of the DL model. The number of nodes chosen varied from 25% to 50% from one study to another. Weight decay also increases the generalizability of the model by regularizing the updated weights, multiplying them by a factor slightly smaller than one; it also reduces the complexity of the model. Dropout and weight were employed in several of the included studies [61,62,65,69,70].

Batch Normalization: This is a technique for training DL models, which standardizes the input to a layer for each mini-batch. It speeds up the training process and increases the performance and generalizability of the models.

Early stopping: This involves stopping the training process at a former point. It helps in determining the number of iterations needed for training the model, before being critically overfitted. It was used in several of the included studies [62,65]. In [65], the validation set was only used to define the early stopping time-point.

Cross-Validation: In this technique, the database is divided into training and testing sets many times by keeping the same proportion but rotating the instances every time. It is a statistical method for testing the performance of classifiers. The most-used cross-validation method is k-fold. In our review, the value of k ranges between 5 and 10. It was used in 85% of included articles

Another major issue that was taken care of by the researchers is data leakage, which also leads to the overfitting and biased evaluation of classification algorithms. It occurs because of using test data in the training process [10].The following four main causes of data leakage were noticed.

Wrong data split: This implies that images of similar subjects are used at several points in time (training, validation, and testing) [154]. For unbiased evaluation or to reduce overfitting, the researchers have to split the data at the subject level, not at the image level.

Late Split: Perform data augmentation after splitting the datasets for an unbiased evaluation. If augmentation is conducted before the split, the same images may be found at multiple points in time.

Prejudice Transfer Learning: When the source and destination targets overlap, when a model is trained during the task of AD vs. NC classification and used for initializing the weights of another task MCI vs. AD, in this case, AD subjects used in the training set of the source task can be in the testing or validation set of the target task. Researchers have to use different datasets for the source and destination, as the authors did in [59,60,65,66].

Absence of an independent validation set: An unrelated validation set (from the test set) should be used for hyper parameter optimization.

### 3.8. Answer to RQ8

Although DL methods with the TL approach have shown better results, the following points still need to be taken care of for developing a clinically acceptable system. The progression detection of MCI to AD is more significant when compared to others for subjects and their loved ones, for future planning, and also for doctors to target those patients that should undergo suitable treatments. As can be seen from Table 7 and Table 8, the maximum accuracy achieved for pMCI vs. sMCI is 87.78% to date, which needs to be improved. Table 7 shows that multimodality performs better than a single modality. The new combination of existing biomarkers and unexplored biomarkers such as tau-PET can be used in the future for predicting the progression of AD.

Despite applying augmentation and TL to avoiding overfitting, the non-availability of large neuroimaging datasets causes generalizability issues. Neuroimaging data synthesis can be employed by researchers in the future to produce new images from prevailing images by using GAN and AE. Still, the viability of synthesis neuroimaging data has to be proven. Owing to the complexity of DL models and 3D neuroimaging biomarkers, it is problematic to display which precise features have been extracted and also to regulate how those features lead to the inference and comparative prominence of specific features [155]. This makes it complex to correct any prejudices that arise from the input datasets. Filter visualization and activation maps [156] were used in [65] to understand the features that contributed significantly to determining the output. It is a challenge to explore the importance of the anatomy of neuroimaging data in predicting the progression of AD for implementing a transparent DL model. The changeableness of configuration and the fortuity involved in training make it challenging to replicate the study and attain identical results. In the future, researchers must reproduce important findings from DL and TL approaches for entirely independent datasets.

## 4. Conclusions

Alzheimer’s disease is one of the leading reasons for death, specifically in developed countries. The timely identification of AD is a demanding task. Therefore, computer-based systems are needed to support physicians in accurate disease classification. DL with TL approaches have been employed in recent years to solve this problem. We started this review with the concept of AD, MCI, tailed by the discussion of existing standards for diagnosis of AD by using MRI, fMRI and PET neuroimaging biomarkers. These biomarkers can be used with DL and TL methods and can be added as an additional tool with some other factors, such as genetic data, MMSE, and Mini-Cog Test results, for better classification. Minimum pre-processing (intensity normalization, registration) is suggested for neuroimaging modalities. Multi-modality can be used for better feature extraction. CNNs with different TL approaches were used in most of the studies. CNN-based DL architectures such as CaffeNet or GoogleNet can be combined with pre-trained models such as ImageNet, VGG16, or inception v4 for implementing more accurate systems. ROI- and patch-based methods are reliable due to their ability to collect relevant information from a brain scan. In addition, GAN may be used for producing novel images from prevailing images for data augmentation. Nevertheless, some challenges such as overfitting concerning the use of small datasets and reproducibility regarding the randomness involved in the training need to be resolved soon. Moreover, it is difficult to determine which features are more significant for an accurate decision support system. This research presents a comparative analysis of the state of the art on DL and TL methods and neuroimaging biomarkers for the early detection of AD. The authors have also analyzed the different input management and pre-processing techniques of neuroimaging biomarkers, different sources of neuroimaging datasets, and the limitations that have been discussed.

State-of-the-art techniques used by researchers for handling overfitting and reproducibility issues have been discussed, along with the software platforms needed to develop DL and TL methods. The pre-processing of biomarkers has also been analyzed. The results show that due to the large size of 3D data, pure DL models took a lot of time in training, as weight initialization was performed randomly. In studies that reported good accuracy, researchers applied only local TL or the combination of pre-trained 3D CNN networks with local TL. These results can be further improved by identifying the most effective combinations of DL and TL with the appropriate biomarkers by conducting continuous experimentation. We identified that the maximum accuracy achieved to date for the early detection of AD is 87.78%, which should be improved in future research. Around 31% of studies used sMRI, while 38% used T1wMRI and other biomarkers such as rs-fMRI,18F-FDG-PET, and AV45-PET. We did not find any implementations of DL with TL that use tau-PET or amyloid-PET. In total, 77% of the studies used voxel and slice-based input management approaches for handling neuroimaging biomarkers, and only 23% of studies used ROI- and patch-based approaches. Around 50% of studies used intensity normalization and registration pre-processing methods, while only 8% of the studies used the biomarkers without any pre-processing methods. The CAFFE framework was used by 30% of the studies for implementing DL architectures, and SPM12 was used by 34% of the studies for the pre-processing of biomarkers,68% studies used ADNI datasets for training as well as testing purposes, only 8% of the studies used their local datasets in combination with ADNI. Future researchers must use different datasets for training and testing to develop a reproducible system.

We hope that this paper will help researchers to obtain a reliable overview of the state of the art for developing a clinically acceptable system for the early detection of AD.

## Figures and Tables

**Figure 1 sensors-21-07259-f001:**
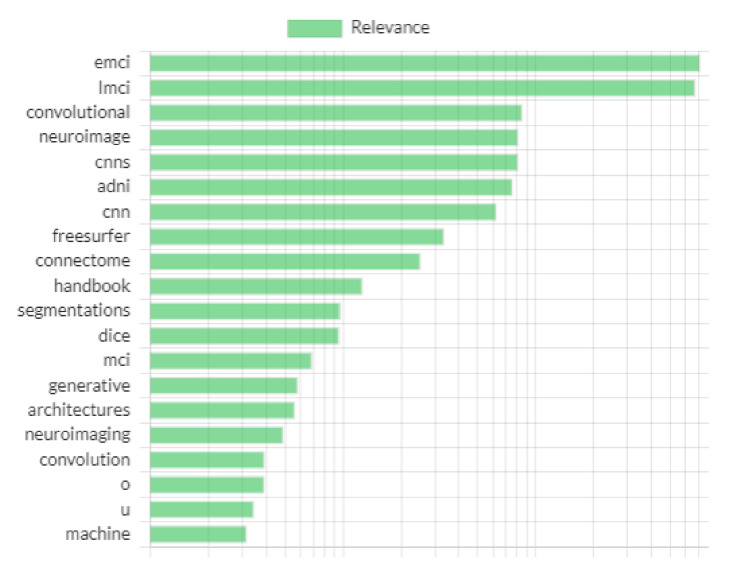
Relevance of important terms identified by Syserv [22].

**Figure 2 sensors-21-07259-f002:**
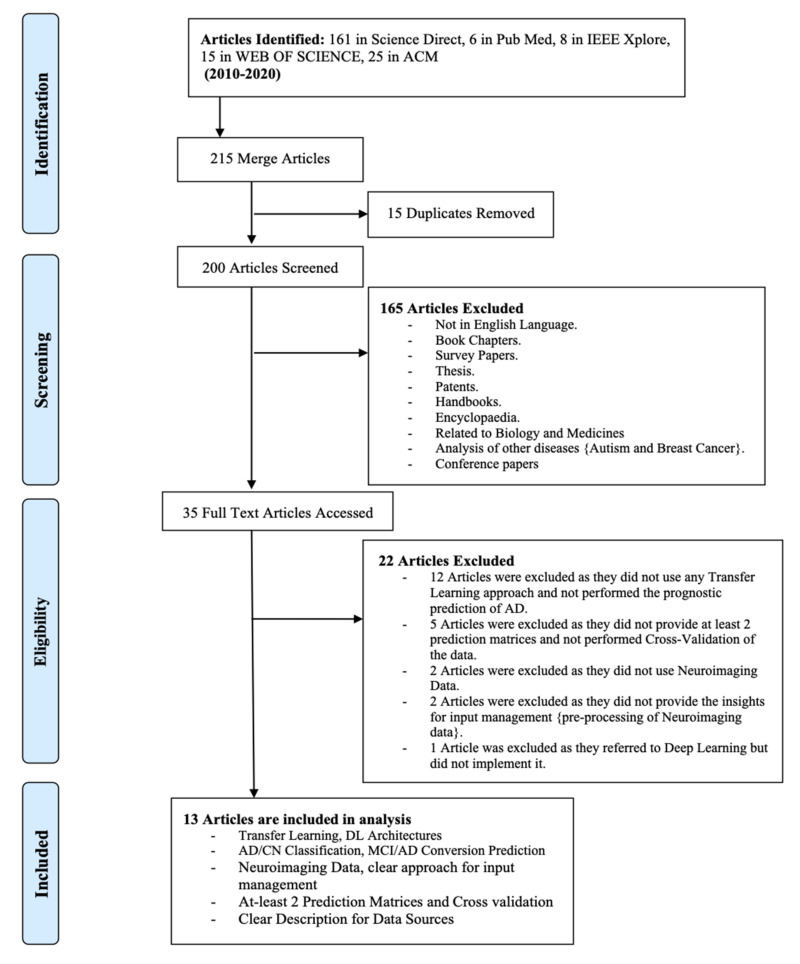
PRISMA diagram of the proposed systematic review.

**Figure 3 sensors-21-07259-f003:**
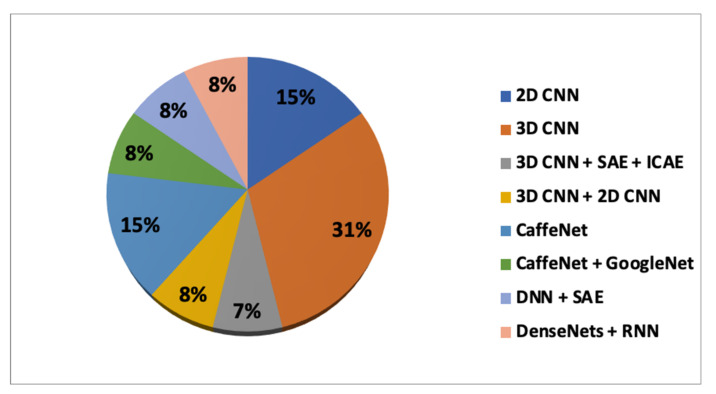
Dissemination of DL architectures in the included studies.

**Figure 4 sensors-21-07259-f004:**
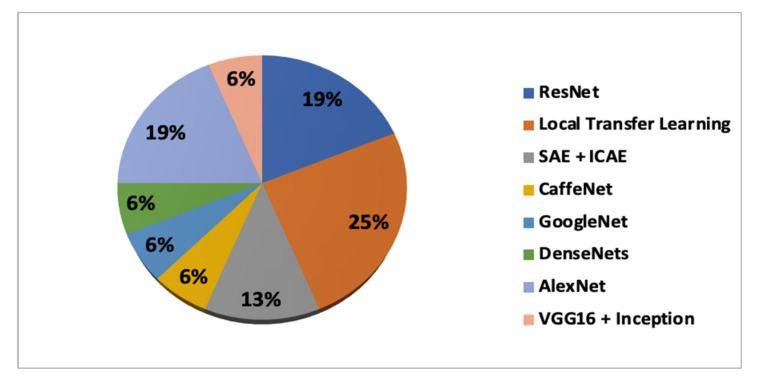
Distribution of TL methods in the included studies.

**Figure 5 sensors-21-07259-f005:**
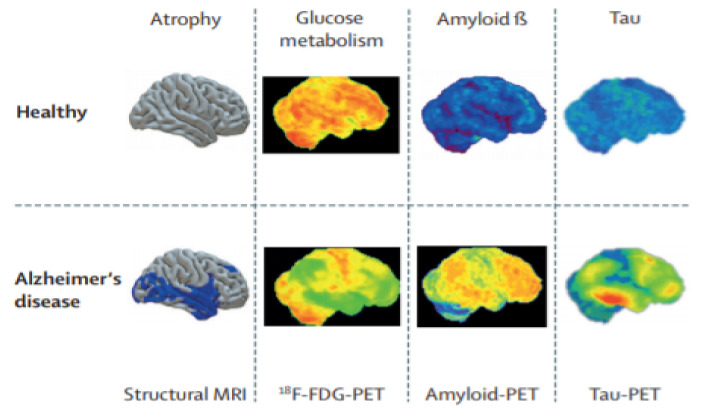
MRI and PET using different radiotracers. Reprinted with permission from ref [91] Copyright 2021 Elsevier.

**Table 1 sensors-21-07259-t001:** Comparison of research for the early detection of AD by using ML, DLandhybrid methods.

Approach	Method	Accuracy of AD vs. NC	Accuracy of sMCI vs. pMCI	Reference
ML Only	SVM (RBF)	81%	70%	[23]
	SVM(LINEAR)	88.9%	70.7%	[24]
	TS-SVM	-	82.5%	[25]
	SVM(LINEAR)	93.1%	75%	[26]
	(GROUP LASSO) SVM	95.1%	65.4%	[27]
	SVM (RBF)	82.5%	69.23%	[28]
DL Only	3D CNN	92.87%	76.21%	[29]
	DBN	90%	78%	[30]
	DNN	84.6%	82.4%	[31]
	3DCNN	91.09%	76.9%	[32]
Hybrid Methods	SAE + SVM	98.8%	83.3%	[33]
	DBM + SVM	95.35%	75.92%	[34]
	RBM+SVM	91.4%	57.4%	[35]

Abbreviations: SVM—State Vector Machine; RBF—Radial Basis Function; TS-SVM—Temporally Structured SVM; DBM—Deep Boltzmann Machine; DNN—Deep Neural Network.

**Table 2 sensors-21-07259-t002:** Distribution of neuroimaging modalities.

Modalities	Percentage	References	N
rs-fMRI	7%	[97,98,99]	1
sMRI	31%	[62,63,64,68]	4
3T T1wMRI	23%	[59,69,70]	3
1.5T T1wMRI	15%	[60,61]	2
18-F FDG PET	8%	[67]	1
18-F FDG PET + sMRI	8%	[65]	1
18-F FDG PET + AV-45 PET	8%	[66]	1

**Table 3 sensors-21-07259-t003:** Accuracy achieved by the type of neuroimaging biomarkers.

Type of Biomarker	AD vs. NC	pMCI vs. sMCI	References
rs-fMRI	97.92%	-	[58]
sMRI	92.78%	77.85%	[62,63,64,68]
3T T1wMRI	94.70%	75.16%	[59,69,70]
1.5T T1wMRI	87.30%	75.97%	[60,61]
18-F FDG PET	-	72.19%	[67]
18-F FDG PET + sMRI	93.58%	82.51%	[65]
18-F FDG PET + AV-45 PET	96.10%	84.20%	[66]

**Table 4 sensors-21-07259-t004:** Distribution of pre-processing methods.

Pre-Processing Method	Used %	References	N
Intensity Normalization	54%	[43,45,46,48,50,54,55]	7
Spatial Smoothing	23%	[58,62,64]	3
Registration	46%	[58,59,60,61,65,67]	6
Gradwrap	8%	[63]	1
Brain Extraction	23%	[58,61,69]	3
Tissue Segmentation	38%	[59,62,64,65,67]	5
High Pass Filtering	8%	[58]	1
Motion Correction	8%	[58]	1
Bias-Field Correction	23%	[60,61,63]	3
Entropy-Based Sorting	8%	[66]	1
No Pre-processing	8%	[68]	1

**Table 5 sensors-21-07259-t005:** Dissemination of input management methods.

Input Management Approach	Percentage	References	N
Voxel-Based	38%	[59,60,64,66,70]	5
Slice-Based	31%	[58,63,67,68]	4
Patch-Based + ROI Based	15%	[65,69]	2
ROI Based	8%	[62]	1
Voxel-based + ROI-based + Patch-based + Slice-based	8%	[61]	1

**Table 6 sensors-21-07259-t006:** The accuracy achieved according to the DL and TL combinations.

DL and TL Combination	AD vs. NC	pMCI vs. sMCI	Ref.
2D CNN + ResNet-18	97.92%	-	[58]
3D CNN + Local Transfer Learning	98.20%	74.90%	[59]
3D CNN + CAE + Local Transfer Learning	85.24% ± 3.97%	73.23% ± 4.21%	[60]
3D CNN + ICAE + Local Transfer Learning	86.60% ± 3.66%	73.95% ± 4.82%	
3D CNN + Local Transfer Learning	88.00% ± 3.00%	78.00%±7.00%	[61]
2DCNN + Local Transfer Learning	79.00% ± 4.00%	-	
3D CNN + Deep ResNet	91.30%	77.80%	[62]
CaffeNet + ImageNet	-	87.78%	[63]
GoogleNet + ImageNet	-	83.23%	
2D CNN + CaffeNet	-	77.98%	[64]
DNN +SAE + Local Transfer Learning	93.58%	82.50%	[65]
Deep CNN + Local Transfer Learning	96.00%	84.20%	[66]
AlexNet + Local Transfer Learning	-	72.19%	[67]
VGG16(from scratch)	74.12%	-	[68]
VGG16 +AlexNet(ImageNet)	92.30%	-	
Inception V4 +ALexNet(ImageNet)	96.25%	-	
DenseNets + RNN+ Local Transfer Learning	89.10%	72.50%	[69]
3D CNN + local Transfer Learning	-	76.00%	[70]

**Table 7 sensors-21-07259-t007:** The accuracy achieved according to the neuroimaging data and input management approach.

Input Management with Biomarker	AD vs. NC	pMCI vs. sMCI	References
Slice-based + rs-fMRI	97.92%	-	[58]
Voxel-based + 3T T1wMRI	98.20%	74.90%	[59]
Voxel-based + 1.5T T1w MP-RAGE MRI	86.60% ± 3.66%	73.95% ± 4.82%	[60]
Voxel-based + T1wMRI	85.00% ± 4.00%	73.00%± 5.00%	[61]
ROI-based + T1wMRI	88.00% ± 3.00%	78.00%±7.00%	
Patch-based + T1wMRI	83.00% ± 2.00%	77.00%±4.00%	
Slice-based + T1wMRI	79.00% ± 4.00%	-	
ROI-based + sMRI	91.30%	77.80%	[62]
Slice-based + sMRI	-	87.78%	[63]
Voxel-based +sMRI	-	77.98%	[64]
ROI-based + Patch-based + sMRI + 18FDG-PET	93.58%	82.50%	[65]
Voxel-based +18F FDG-PET + AV-45 PET	96.00%	84.20%	[66]
Slice-based + 18F FDG-PET	-	72.19%	[67]
Slice-based + sMRI	96.25%	-	[68]
ROI-based + Patch-based + 3T T1wMRI	89.10%	72.50%	[69]
Voxel-based + 3T T1wMRI		76.00%	[70]

**Table 8 sensors-21-07259-t008:** Distribution of datasets.

Datasets	Used Percentage	Reference	N
ADNI	69.23%	[43,45,47,48,49,50,51,52,54]	9
OASIS	7.69%	[68]	1
IXI + ADNI	7.69%	[70]	1
OASIS + ADNI + AIBL	7.69%	[60]	1
Local (MILAN) + ADNI	7.69%	[59]	1

**Table 9 sensors-21-07259-t009:** Distribution of pre-processing software.

Pre-Processing Software	Percentage	References	N
FSL	17%	[58,69]	2
SPM12	34%	[59,60,62,64]	4
MATLAB	17%	[63,68]	2
FreeSurfer	8%	[65]	1
NifTi_2014 toolkit	8%	[67]	1
Nipype	8%	[61]	1
MRIcron	8%	[70]	1

**Table 10 sensors-21-07259-t010:** Software packages used for implementing DL and TL algorithms.

Ref	Software Architectures Used for Implementing DL and TL Algorithms
[58]	Implemented on CAFFE Framework and trained on theFloydHub cloud service with GPU Tesla K80.
[59]	Python using Theano by using Dell Powerdge T630 Linux, including high-performance GPU NVIDIA Tesla K40, with 2880 CUDA cores and High-Frequency Intel Xeon E5 –2623 v3 with 78 GB memory overall.
[60]	Not mentioned
[61]	The DL models were built using the Pytorch library 5. TensorboardX 6 was embedded into the current framework to dynamically monitor the training process.
[62]	Training and testing routines were implemented on an NVIDIA CUDA parallel computing platform using GPU-accelerated CUDA toolkit/compilation and Pytorch python package tensor libraries
[63]	CAFFE Framework with NVIDIA DIGITS and NVIDIA Quadro M4000 (GPU).
[64]	Not mentioned
[65]	Open-source Deep Learning toolbox
[66]	MatConvNet
[67]	CAFFE Framework
[68]	Keras with TensorFlow backend
[69]	PC with Ubuntu14.04-x64/GPU NVIDIA GTX1080Ti of 11GB memory, python 2.7.9, Keras library with the Tensorflow backend.
[70]	Not mentioned

**Table 11 sensors-21-07259-t011:** Dissemination of software packages used for implementing DL models.

Software Packages	Percentage	References	N
CAFFE framework	30%	[58,63,67]	3
Tensorflow + Keras	20%	[68,69]	2
Pytorch	20%	[61,62]	2
Deep Learning Tool-Box	10%	[65]	1
Theano	10%	[59]	1
MatConvNet	10%	[66]	1

## Data Availability

Not applicable.

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
