# Peer review of "Transfer Learning for Alzheimer’s Disease through Neuroimaging Biomarkers: A Systematic Review"

_sensors, 2021, doi:10.3390/s21217259_

Round 1
Reviewer 1 Report
Too many abbreviations in the Abstract, making it difficult to understand at first reading. This continues in the introduction chapter, where the authors in the first paragraph use twenty abbreviations, including repetitions. Authors should not abuse the use of abbreviations, applying them only when it is very repetitive.
Figure 1 is not understood. Excessive use of abbreviations (AD, NC, pMCI, sMCI), use of undefined expressions in the caption (Task-1, Task-2), two rectangular zones with the same text (Image-Pre- Processing...). Suggestion: remove the figure!
The paper written by the authors does not follow the traditional structure of a review paper in the engineering field, as this paper includes research questions and reflection for the answers, so this work can be classified as a research work. In this line of thought, it is suggested to remove A systematic review from the title, being replaced by a more appropriate expression, eg. gaps, challenges, and opportunities.
In section 2.2 the authors do not justify the choice of the places where they did the research, and that may result in works found in one place also existing in another place. They also do not explain what they mean by articles, if are just papers from international journals or if also include works published in conferences.
Table 1 can be replaced by plain text, as it does not have relevant information to appear tabulated.
Confusing use of the word article, replace with work.
A work that analyzed in detail only 13 works, out of a set of 215 initial works, the authors should repeat the research, increasing the range of initial works. Again, a study based on 13 works is very limiting.
The authors created research questions (RQ) and in the results chapter they seek answers for each of the RQs. Despite being debatable, in my opinion (given the review articles published in international journals) the authors should re-write the text of the paper, not defining RQ and searching the answers, but using each RQ as the title of a subsection taking advantage of the development text already written.
WARNING: do not use images already published in other papers, as copyright infringement may occur (see figure 5 with ref. [76] )
With such extensive work, the conclusions are too compact not showing the value of the current work. Authors should increase the conclusions chapter by including more details on what they concluded for each RQ, in order to enhance the present work.
ALSO:
- Tables 6 and 7 are together with no text between them. The same happens with tables 10 and 11.
- Some references are incomplete (eg [2], [12], [89]): lack of publication date or reading date.
- The authors use $305 billion without caps so $1.1 Trillion should be also without caps.
- Replace doctors with physicians.
Reviewer 2 Report
The paper presents a state-of-the-art in transfer-learning based recognition of Alzheimer’s disease from various imaging modalities. Such review is very important as it collects and discusses most recent and advanced research in the field. This review, however, requires some improvements before it will be suitable for publication.
- Literature analysis limited to 11 publications only is too narrow to be named “A systematic review”. I recommend softening exclusion criteria and consider much more papers on this topic. For example, it is not clear why book chapters and conference papers were totally excluded since importance of some conferences (and paper rejection rates) are much higher than of some journals.
- The most important information about methods efficiency was moved to supplementary materials (actually, there is only one table in appendix E that compares different approaches in terms of efficiency). Such information should be provided in main text while other information (like detailed discussion about percentage of different network architectures implemented for AD detection) should be moved appendices.
- There is a lack of discussion of results obtained by presented systems. The reader would like to know which method is most promising, what are advantages and disadvantages of discussed approaches, why some network architectures perform better than other on similar data. Also, it should be discussed which imaging method (and why) is most suitable for AD analysis from the point of view of algorithms applied.
- Also, comparison with other approaches should be provided. For example, not transfer learning deep network should be briefly analysed as well as classical algorithms (no deep learning based). Such analysis and efficiency comparison would of great interest for readers.
Round 2
Reviewer 1 Report
The authors made minor changes to the work. Most responses to the reviewer are convincing and demonstrate trustworthiness in the subject.
The new version of the work has table 5, which occupies 1 page. But discussion about this table is almost non-existent. Given that most of the cells in this table are empty and as the authors do not discuss the results comparing values, justifying with scientific features the differences in values, the need for this table is questionable, at least in the present form. Recommendations for authors: (i) only some values have the corresponding error (example Ref [45] for AD vs NC), so it is preferable to remove the error; discuss the values presented, explaining why some values are higher than others, what is different in the compared methods that can justify the differences; column widths must be constant and not some columns wider than others; if it is impossible to include a scientific discussion of the results presented in this table, it may be removed, as the authors already discuss the values shown in the other tables.
Additional details:
- In the listing of references, include the title or its meaning in references [118] and [119]
- Some details to modify for the final version of the document: font sizes in figure 3 are too large (but in figure 4, they are already OK).
Author Response
"Please see the attachment."

Reviewer 2 Report
Thank you for addressing the points raised in my review. However, two matters require further clarification or explanation.
- I did not receive an answer why publications in the form of chapters in books or conference materials were not included in the review. The selection of only about 5% (11 out of over 200) publications on AD detection using DL/TL was not justified. Again, please provide convincing arguments for such strict exclusion criteria or include more papers in the review.
- I do not require such detailed analysis of other AD detection methods as these presented in this study (with use of DL/TL). However, when focusing on such methods, it is necessary to relate them (even in a limited way) to others (‘classic’ machine learning, DL without TL). Please show some examples of other AD detection methods and compare their effectiveness in relation to the analyzed methods. Currently, you provided no evidence that the AI detection methods using DL/TL are superior to others. For this reason, the reader will feel confused and will not understand why this particular type of method has been discussed in detail.
Author Response
"Please see the attachment."
